# Unravelling the Genomic and Virulence Diversity of *Legionella pneumophila* Strains Isolated from Anthropogenic Water Systems

**DOI:** 10.3390/microorganisms13122832

**Published:** 2025-12-12

**Authors:** Sofia Barigelli, Piotr Koper, Maya Petricciuolo, Andrea Firrincieli, Marta Palusińska-Szysz, Ermanno Federici

**Affiliations:** 1Laboratory of Applied and Environmental Microbiology, Department of Chemistry, Biology and Biotechnology, University of Perugia, 06122 Perugia, Italy; sofia.barigelli@dottorandi.unipg.it (S.B.); maya.petricciuolo@unipg.it (M.P.); 2Department of Genetics and Microbiology, Institute of Biological Sciences, Maria Curie-Skłodowska University, 20-033 Lublin, Poland; piotr.koper@mail.umcs.pl (P.K.); marta.palusinska-szysz@mail.umcs.pl (M.P.-S.); 3Bioinformatics and Biostatistics Laboratory, Institute of Biological Sciences, Maria Curie-Skłodowska University, 20-033 Lublin, Poland; 4Department for Innovation in Biological, Agro-Food and Forest Systems, University of Tuscia, 01100 Viterbo, Italy; andrea.firrincieli@unitus.it

**Keywords:** *Legionella pneumophila*, environmental strains, SBT, cgMLST, pangenome, virulence factors, cytotoxicity, intracellular replication

## Abstract

*Legionella pneumophila*, a waterborne pathogen naturally present in freshwater and capable of colonizing artificial water systems, is responsible for Legionnaires’ disease (LD)*,* a severe form of pneumonia transmitted through inhalation of contaminated aerosols. Virulence of *Legionella* strains is affected by the plasticity of their genome, shaped by horizontal gene transfer and recombination events. Thus, contaminated water systems can host diverse *Legionella* populations with a distinct virulence potential. Here, we compare the genomic diversity of *Legionella pneumophila* strains isolated in water systems of academic buildings, together with their cytotoxicity and intracellular replication in THP-1-like macrophages. A six-year environmental surveillance revealed *Legionella pneumophila* contamination in 20 out of the 50 monitored sites, identifying five serogroups (sg) and 13 Sequence Types (STs). Phylogenetic investigations based on core genome multilocus sequence typing (cgMLST) and comparative genomics of representative isolates of each ST showed a broad diversity and a heterogeneous virulence repertoire, especially within the Dot/Icm and Lvh secretion systems. Following macrophage infection, a strain-dependent cytotoxicity and intracellular replication was observed, underlying significant pathogenic diversity within the same species and stage-dependent infection dynamics. Together, these results showed strain-specific genetic and phenotypic virulence traits to be considered during risk assessment in environmental surveillance.

## 1. Introduction

The waterborne bacterium *Legionella pneumophila* (*L. pneumophila*) is the primary etiological agent of Legionellosis, a respiratory syndrome that can result in an atypical and potentially fatal form of pneumonia, Legionnaires’ disease (LD) [1]. Clinical cases have been observed worldwide, with an increase in incidence of the disease in Europe, the U.S., Canada, and Australia [2]. In the last decade, Italy was one of the European countries with the highest number of LD notification cases, with 4.7 cases per 100,000 population reported in 2019, 4.6 cases per 100,000 in 2021, and a further increase of 13% was registered in 2022. In the same year, the case-fatality rate for community-acquired LD was 15.1%, compared to the European Union/European Economic Area (EU/EEA) average of 9% [3,4,5]. In 2023, 66.3 cases per million inhabitants were registered, with an increase compared to the previous year (52.8/1,000,000) [6].

LD symptoms develop with the replication of *Legionella* in human alveolar macrophages after inhalation of contaminated aerosols. Artificial water systems, including drinking water distribution systems, cooling towers, air-conditioning units, bathtubs, and showers, are considered the primary source of infection [7,8]. In water systems, *Legionella* can actively proliferate between 25 °C and 45 °C, even if the pathogen can also persist across a wider thermal range [9]. Although *L. pneumophila* can survive as free-living cells, the colonization of biofilms and infection of protozoa enhance its persistence in the environment by providing nutrients and resistance to antibiotics and disinfectants [10]. Although most of the reported LD cases and outbreaks have been linked to hospitals, nursing homes, hotels, and residential complexes, occupational settings such as schools and universities may be regarded as potential at-risk facilities. For instance, high contamination levels were detected in academic campuses, requiring subsequent corrective disinfection actions [11]. Therefore, the need for monitoring *Legionella* contamination has increased alongside the presence of more anthropogenic reservoirs due to a higher risk associated with exposure to *Legionella*-containing aerosols [12,13].

European and Italian guidelines [14,15] aim to prevent and control the risk of *Legionella* proliferation and transmission by providing technical and management measures to safeguard public health. Furthermore, the ISO 11731:2017 legislation [16] presents a culture-dependent approach as the gold standard for *Legionella* enumeration and identification. Contamination thresholds, established accordingly to this method, have been defined as reference values to guide corrective actions when exceeded. *Legionella* typing is considered a useful approach in epidemiological investigations and, more recently, for environmental surveillance programs, as the distribution of *Legionella* strains varies across man-made water systems, such as those of hospitals, residential and occupational buildings, and hotels [17,18,19]. In this respect, molecular typing has been used as a support for *L. pneumophila* environmental surveillance. Sequence based typing (SBT) is the gold standard for strain typing [20,21], as it enables the study of genetic diversity and clonal population structure, as well as the comparison of clinical and environmental isolates in epidemiological investigations. Over the last few years, the SBT scheme, based on seven *Legionella*-specific genes, has been increasingly complemented by the more discriminatory technique of Whole Genome Sequencing (WGS). The information gained from the entire genome of *Legionella* provided support for molecular epidemiology in the investigations of environmental contaminations and the identification of infection sources [22,23].

The more extensive use of WGS brought insights regarding the genomic characteristics of *Legionella* members. *Legionella* genomes are highly diverse in both size and content. The core genome (cg) comprises only 1008 genes, underscoring the genomic diversity of the genus, and the pan-genome is far from being fully described [24]. Studies focused on the pangenomics of the species *L. pneumophila* revealed the presence of a core of 1979 genes, 342 strain-specific genes, and 18 strain-specific genomic islands [25]. Moreover, WGS comparisons showed that recombination events are responsible for the genetic diversity of the species and shape 34% to 57% of the *L. pneumophila* genome. Indeed, the exchange of a substantial portion of genomic regions between strains has been demonstrated (e.g., between Philadelphia and Paris strains) [26]. Genomic differences observed within the *Legionella* genus are also the result of Horizontal Gene Transfer (HGT) among *Legionella* and prokaryotic and eukaryotic microorganisms, alongside the presence of mobile genetic elements within the accessory genome, as recently reviewed [27].

The ability of *Legionella* to replicate in macrophages is responsible for the manifestation of the disease. The bacterium switches from a transmissive to a replicative form once in the host, within a *Legionella*-containing vacuole (LCV). In the intracellular environment, the pathogen subverts the endocytic and metabolic pathways of the eukaryotic cell to grow. *Legionella* steps back into the transmissive form upon nutrient limitations and egresses from the macrophage, inducing its necrosis [28]. Environmental strains of the genus *Legionella* have shown differences in infecting and replicating within macrophages, as well as in causing pathogenic consequences for their hosts. Hence, virulence phenotypes could be genotype-related and based on their virulence gene patterns [12,29,30]. The exchange of genetic material plays a role in defining the virulence potential of species and strains, particularly when it involves genes encoding effectors, which are implicated in the intracellular replication of *Legionella* in human macrophages [31,32].

Here, we investigated the diversity of *L. pneumophila* contaminations in drinking water systems and a cooling tower of academic buildings located in a single city, by serogroup identification and sequence-based typing. To further explore the genomic relationships and virulence gene repertoires of the *Legionella* strains collected, we selected 13 isolates, each representing a different ST retrieved from the contaminated samples, for WGS analysis. In addition, we implemented in vitro infection assays with THP-like macrophages to assess the functional features of the strains beyond their genomic patterns. In particular, we assessed the ability of the environmental strains to induce host cell death, which occurs at the end of the intracellular growth cycle of *Legionella* [33], and their intracellular replication ability.

## 2. Materials and Methods

### 2.1. Water Sampling

From 2015 to 2021, 50 sites belonging to the University of Perugia, Italy, namely 49 buildings and one cooling tower, were monitored. Sampling was performed following the Italian Guidelines [15]. After flushing water for 2 min, and until the temperature stabilized, one liter was collected from the hot water faucets using 0.1% sodium thiosulfate containers to neutralize residual chlorine. Samples from a Heating, Ventilation and Air Conditioning (HVAC) cooling tower, maintained under chlorine-based treatment according to European and Italian guidelines [14,15], were also collected, by taking one liter directly from the cold water basin. All water samples were processed within 24 h.

### 2.2. Isolation and Enumeration of Legionella in Samples Collected from Drinking Water Systems and Cooling Tower

*Legionella* presence was assessed in compliance with ISO 11731:2017. Bacteria collected on a 0.22 µm pore polycarbonate membrane (Merck Millipore, Billerica, MA, USA) after the filtration of one liter of water were resuspended in 10 mL of the original sample. Then, 0.1 mL of the suspension was spread onto Buffered charcoal yeast extract agar (BCYE) and Glycin vancomycin polymyxin cycloheximide agar (GVPC) (Thermo Fisher Scientific, Waltham, MA, USA), which were incubated at 36 ± 1 °C for 10 days. At least three representative colonies displaying *Legionella* morphology from each plate were sub-cultured on BCYE with and without cysteine (Thermo Fisher Scientific, Waltham, MA, USA) at 36 ± 1 °C for 48 h. Colonies were confirmed as *Legionella* when able to grow in the presence of cysteine but not in its absence. The number of *Legionella* colony-forming units per liter (CFU/L) in the original water samples was then calculated from the plates with the highest number of confirmed colonies, considering the dilution performed.

### 2.3. Phenotypical and Genotypical Typing of Legionella Isolates

The serogroup (sg) of the isolates confirmed to belong to the *Legionella* genus was assessed by using the *Legionella* Latex Test (Thermo Fisher Scientific, Waltham, MA, USA) to discriminate among *L. pneumophila* sg 1, *L. pneumophila* sg 2-14, or *L. non-pneumophila*. Isolates that showed agglutination for *L. pneumophila* sg 2-14 antibodies were further distinguished into specific serogroups with the *Legionella pneumophila* antisera sets (Biogenetics, Padova, Italy). Isolates positive for the *L. pneumophila* Latex Test were characterized by SBT using the seven *loci* standard protocol established by the European Study Group for *Legionella* Infections (ESGLI). After DNA genomic extraction by thermal shock method, *flaA*, *pilE*, *asd*, *mip*, *mompS*, *proA*, and *neuA/neuAh* genes were amplified [20,21,34]. The PCR mix contained FIREPol 5× Master Mix (Solis BioDyne, Tartu, Estonia), 0.4 μM forward and reverse primers [20], and 4 μL of template DNA in a total volume of 20 μL. The amplification protocol consisted of 34 cycles of denaturation at 95 °C for 20 seconds, annealing at 60 °C for 30 seconds, and extension at 72 °C for 1 min. Amplicons were purified using the EuroSAP PCR Enzymatic Clean-up kit (Euroclone SpA, Pero, MI, Italy) and sequenced (Eurofins Genomics, Ebersberg, Germany). The allelic profile of each *L. pneumophila* strain was assessed with the European Working Group for *Legionella* Infections (EWGLI) SBT database (Public Health England). A minimum spanning tree (MST) was built using the goeBURST algorithm implemented in the PHYLOViZ Online software version 2.0 (https://online.phyloviz.net/index, accessed on 6 August 2025). Sequence types (STs) sharing five or more *loci* were considered part of a clonal complex (CC); otherwise, they were called singletons [35].

### 2.4. DNA Extraction and Whole Genome Sequencing

One representative isolate of each strain identified by SBT was selected to perform WGS. The strains were cultured on BCYE agar at 37 °C for 48 h, and genomic DNA extraction was performed following the GenEluteTM Bacterial Genomic DNA Kit specifications for Gram-negative bacteria (Sigma-Aldrich, St. Louis, MO, USA). Sequencing was performed using NovaSeq Illumina platform (Eurofins Genomics, Ebersberg, Germany), obtaining pair-end 150 bp reads. Data were processed and analyzed using a High-Performance Computing platform [36]. The quality of the sequences was checked with FastQC (version 0.12.1;) while BBTools (v39.28) [37] was used for quality trimming, adapter removal, and sequencing coverage normalization. High-quality reads were de novo assembled with Unicycler (v0.5.1) [38] with default settings. Quality and contamination levels assessments of assemblies were carried out through QUAST (version 5.3.0) [39] and CheckM2 (version 2.4.0) [40]. The core genome Multilocus Sequence Typing (cgMLST) scheme of Moran-Gilad [41], characterized by 1521 core genes, was converted using the chewBBACA software (v3.3.10) [42]. The allelic profiles, including the genes present in all the strains, were used to build an MST with the PHYLOViZ Online software version 2.0 (https://online.phyloviz.net/index, accessed on 6 August 2025). STs were considered as part of the same cluster when differing from no more than four *loci* [41]. Pangenome analysis was performed by following the anvi’o version 8 pangenome workflow [43]. Briefly, all the genomic FASTA files were converted into an anvi’o contigs database for gene calling and functional annotation through Clusters of Orthologous Groups (COG) and included in a new anvi’o genomes storage for pangenome analysis. The full pangenome was split into core, soft-core, shell, and cloud bins based on the gene cluster frequency across the genomes. Predicted protein sequences were further analyzed using the Virulence Factor Database (VFDB) VFanalyzer online tool (https://www.mgc.ac.cn/VFs/, accessed 6 August 2025) [44]. The resulting Excel tables were parsed with custom R scripts to construct binary presence/absence matrices for each virulence factor, grouped by functional class. Reference genomes (*L. pneumophila* Paris, Lens, Corby, Philadelphia-1, and NCTC12821) were included for comparison. Heatmaps were generated with the Complex Heatmap package in R, showing only variable virulence factors. Complete presence/absence data are provided in Appendix A.

### 2.5. THP-1-like Macrophage Infection by Legionella pneumophila Strains

Before each infection experiment, exponentially growing THP-1 cells (macrophage cell line provided by Cytosens S.R.L.S. (Napoli, NA, Italy)) were counted and seeded in a 48-well plate at a concentration of 2 × 10^5^ c/well in RMPI 1640 supplemented with 1% pen-strep (Euroclone SpA, Pero, MI, Italy) and 10% FBS (Thermo Fisher Scientific, Waltham, MA, USA). Cells were incubated with phorbol-12-myristate-13-acetate (PMA) (Thermo Fisher Scientific, Waltham, MA, USA) 300 nM at 37 °C in 5% CO_2_ for 72 h to allow for their differentiation. *L. pneumophila* strains were cultured on BCYE medium at 37 °C for 72 h. A reference strain, *L. pneumophila* NCTC 12821 (provided by Dr. Osvalda De Giglio, Interdisciplinary Department of Medicine, Section of Hygiene, University of Bari Aldo Moro, 70124 Bari, Italy) was also used for comparison in infection assays. On the day of the experiment, the bacterial biomass was suspended in sterile PBS, and the absorbance was read at 600 nm. The suspension was diluted to reach an OD of 0.25, which corresponded to 2.2 × 10^8^ c/mL, as previously determined by plate count on BCYE. The suspension was then diluted to the desired multiplicity of infection (MOI). Differentiated macrophages were washed with PBS and infected with the correct MOI in RMPI without antibiotics and FBS. Multi-wells were centrifuged at 270× *g* for 10 min to synchronize the infection. After 2 h of co-incubation at 37 °C and 5% CO_2_, macrophages were washed and treated with gentamicin (Euroclone SpA, Pero, MI, Italy) 100 ng/mL for 1 h at 37 °C and 5% CO_2_ to kill extracellular bacteria. Direct plating of cell culture supernatant on BCYE confirmed the absence of viable *Legionella* cells. At the end of the incubation, the wells were washed again with PBS. For cytotoxicity assays, RMPI without antibiotics and 10% of FBS were added and cells incubated at 37 °C and 5% CO_2_ for 24 and 48 h. Macrophage viability was determined after the addition of 10% Alamar Blue (Euroclone SpA, Pero, MI, Italy) and incubation for 5 h, by measuring absorbance at 550 nm, with each condition tested in triplicate. The mean absorbance values were then compared to those of uninfected control cells to determine the relative percentage of viable macrophages. For intracellular replication assays, cells seeded in the wells used to determine the entrance of *Legionella* strains were lysed with 100 μL of 0.01% Triton-X-100 (Thermo Fisher Scientific, Waltham, MA, USA) and plated on BCYE after serial dilutions. Wells prepared to assess the intracellular replication of *L. pneumophila* strains were filled with RMPI without antibiotics and 10% of FBS. Cells were lysed with 100 μL 0.01% Triton-X-100 after 24 and 48 h of incubation and spread on BCYE plates after serial dilutions. Colonies were counted for each time point after 72 h at 37 °C.

### 2.6. Statistical Analyses

Statistical analyses were performed using R Studio software (2023.09.0 + 463 version). Comparison among results was carried out using a One-way ANOVA with Tukey’s HSD *post hoc* and Student’s *t*-test (*rstatix* package). All analyses were applied at a 95% level of confidence and a level of significance for *p*-values < 0.05. A genotype–phenotype correlation analysis was performed using VFDB-based virulence profiles and phenotype-based data, namely Δlog10 CFU at 24/48 h; macrophage survival at 24/48 h for MOI 1 and 10. Correlations were computed for individual virulence factor genes, as well as for VFDB functional classes, where class-level values were defined as the sum of genes present in a class for each strain. Pearson correlation coefficients were computed for each gene or class versus each phenotype, with Benjamini–Hochberg correction applied within phenotype families.

## 3. Results

### 3.1. Diversity of Legionella pneumophila Contaminations in Drinking-Water Systems and Cooling Tower of Academic Buildings

The surveillance of *Legionella* contamination carried out between 2015 and 2021 indicated that among the 50 sites (49 buildings and one cooling tower) monitored, 20 presented, at least once, a *L. pneumophila* contamination. These contamination cases were ascribed to 13 different STs, as determined by SBT allelic profiles, and 5 different serogroups, as reported in Table 1. Table 1 also provides information about the samples from where the STs were isolated, namely the contaminated water systems, the year of occurrence, the average temperature and the average contamination levels.

Among all of the *L. pneumophila* isolates, five different serogroups were identified. Sg1 was the most abundant (65.96%), followed by sg8 (2.41%), sg6 and sg7 (4.25%), and sg4 (2.13%). The water temperature of the samples from which the isolates originated ranged from 12.8 °C to 61 °C, with contamination levels varying from 100 to 425,000 CFU/L. The SBT analysis allowed us to identify 13 different strains, among which ST1 was the most frequently isolated (40.43%). The other STs that were identified more than once were ST445 (12.77%), ST1324 (10.64%), ST1358 (8.51%), ST356 (6.38%), ST1904 (4.26%), and ST2617 (4.26%). The remaining STs (ST22, ST23, ST579, ST596, ST1541, ST3130) were obtained only once (2.13%). ST3130 was a novel ST that had never been isolated before; therefore, it was submitted to the SBT database, together with the 7 loci sequences, following its identification in our environmental surveillance project.

To unravel the relationships between the strains, an MST based on the SBT profiles was performed using the goeBURST algorithm (Figure 1).

### 3.2. Whole Genome Sequencing and Phylogenomic Analysis of Environmental L. pneumophila Strains

Thirteen environmental *L. pneumophila* isolates, each representing a different ST identified, were selected for WGS analysis to explore their genomic relationships and content. The number of contigs determined by using QUAST varied from 26 to 67 for a genome size which ranged from 3.2 Mb to 3.6 Mb. The average N50 value was 0.2 Mb, while the average GC content was 38.1%. CheckM2 quality analysis determined a maximum of 0.26% of contamination and 100% of completeness (Table 2 and Appendix A).

The phylogenetic relationship between the de novo assembled genomes was assessed through the cgMLST scheme, according to which all the 13 isolates had from 96 to 99% of successful 1521 targets (the cut-off of acceptable data being 95%) [41]. A cgMST was built based on the 1411 gene targets detected in all the genomes (Figure 2).

### 3.3. Pangenome Analysis of L. pneumophila Environmental Strains

To obtain a comprehensive overview of the genomic diversity among the *L. pneumophila* isolates representative of each ST identified, we performed a pangenome analysis. By comparing the distribution of shared and unique genes, this approach sought to highlight genetic variability and provide context for the strain-dependent differences (Figure 3). The anvi’o analysis identified a total of 4506 gene clusters (GCs) and 40,007 gene calls. A total of 2475 GCs (54.93%) belonged to both core (i.e., shared by 99% of the isolates) and soft-core (i.e., shared by 95% of the isolates) bins. Moreover, the 1171 GCs (25.99%) detected in 15% to 95% of the strains were identified as part of the shell bin, while the cloud bin, indicating the group with less than 15% of shared GCs, contained 860 GCs (19.08%). Core (core and soft-core) and accessory (shell and cloud) bins were annotated into Clusters of Orthologous Groups (COG) classes. The most representative GCs of the core genome were related to functions involved in translation and ribosomal biogenesis (category J; 11.85%), amino acid transport and metabolism (category E; 8.18%), and in the biogenesis of the cell wall and membrane structures (category M; 7.52%). Within the accessory genome, GCs were mostly associated with virulence and defense mechanisms (categories U and V; 11.24%), general predicted functions (category R; 11.07%), and signal transduction mechanisms (category T; 7.32%). Among the GCs identified as singletons, virulence COG categories showed a different distribution across the *L. pneumophila* strains. Their frequency was the highest in ST1 (23.5%) and ST356 (17.6%), and the least in ST22 (2%), ST23 (2%), and 2617 (2%).

### 3.4. Virulence Gene Repertoire of Environmental L. pneumophila Strains

The screening of the 13 environmental isolates, together with reference strains (including *L. pneumophila* NCTC 12821, used as a control in infection assays) against the VFDB database revealed that most virulence determinants were highly conserved across the dataset (Figure 4). Core *Legionella* surface-associated factors, including type IV pili, Mip, MOMP, and Hsp60, were detected in all isolates and references, as well as iron uptake systems, regulatory proteins, and the majority of the Dot/Icm type IVB secretion apparatus. In contrast, variation was observed for two secretion modules. For the Dot/Icm (type IVB) system, we identified 39 effector proteins, of which 22 were present in all isolates, whereas 17 showed heterogeneous distributions. The conserved set comprised Rab1-targeting and vacuole-stabilizing proteins (e.g., LidA, LepA/LepB, RalF, SdhA/B), whereas the variable set included effectors such as SidM/DrrA together with its antagonist SidD, members of the Sde family, and VipA/E. The second variable module was the Lvh (type IVA) secretion system. Several STs (e.g., ST1, ST22, ST1324, ST1904, ST2617) carried a complete Lvh *locus*, while others lacked it entirely or retained only fragments. This distribution mirrors reference strains, with Paris, Lens, and Philadelphia-1 encoding a full Lvh system, whereas Corby lacks it. Beyond secretion-related variation, a small number of accessory determinants were annotated sporadically. In particular, VFDB screening suggested the presence of capsule synthesis genes with similarity to *Acinetobacter* and O-linked flagellar glycosylation (pseB) genes resembling those of *Campylobacter*, both detected exclusively in ST596 and ST3130.

### 3.5. Host-Cell Death-Inducing Potential of Environmental L. pneumophila Strains Towards THP-1-like Macrophages

To investigate the potential of the 13 *L. pneumophila* environmental strains to induce host cell death, the survival of THP-1 macrophages was measured 24 and 48 h after exposure to *Legionella* at multiplicities of infection (MOIs) of 1 and 10. Figure 5 illustrates the cytotoxic effect of all strains compared to the control (uninfected cells), with color intensity reflecting the degree of damage caused by each ST. Results are reported for each time point and MOI.

Overall, all the strains infected and killed the eukaryotic hosts with distinct intensities related to the MOI and post-infection time. As expected, the cytotoxic effect increased over time. Indeed, after 24 h, not all strains caused cell mortality, especially at MOI 1. After 48 h, cell death was detected at both MOIs for all strains, as the survival of infected macrophages was always significantly lower than that of the control. Moreover, a tenfold increase in the MOI resulted in a significant rise in the cytotoxicity of the strains. At MOI 10, all strains displayed a higher ability to kill their host at the same time point. Among the *L. pneumophila* strains tested, ST22, ST596, and ST1324, as well as the reference strain, were those with the highest ability to kill macrophages, regardless of the MOI and the time point. Other STs, such as ST1, ST356, ST2617, and ST3130, also showed a similar pattern of cytotoxicity, although to a lesser extent. In contrast, ST23, ST445, ST579, and ST1541 showed a low cell-killing effect, failing to induce significant host death at MOI 1 after 24 h, while ST1904 was the least effective, with significant cytotoxicity observed only after 48 h.

To evaluate the different cytotoxic potential of environmental *L. pneumophila* strains, a comparative analysis of the survival rates of THP-1 macrophages infected with our selected STs was performed (Figure 6). Differences between strains were detected in every experimental condition tested, as displayed in the matrix under each chart. However, the highest number of significant differences was observed at MOI 1 after 48 h of incubation (Figure 6c). At 24 h, strain-dependent differences were detected at both MOIs, although to a lesser extent (Figure 6a,b). Conversely, after 48 h of infection at MOI 10, THP-1 survival rate was drastically reduced, which resulted in the least number of statistically significant differences ever detected (Figure 6d).

Pearson correlation coefficients were computed to search for correlations between macrophage survival and virulence-associated genes and classes (Appendix A, respectively). Although none remained significant after multiple-testing correction, a trend showing a negative association between Lvh (type IVA) system genes and macrophage survival (i.e., higher cytotoxicity) was observed.

### 3.6. Replication Ability of Environmental L. pneumophila Strains Within THP-1-like Macrophages

As an intracellular pathogen, *Legionella* acquires the nutrients of the host cells to replicate. Part of our in vitro infection studies focused on highlighting differences in the entry of *L. pneumophila* strains into macrophages, as indicated by intracellular growth in THP-1 cells at 0 h (T0), and evaluating their replication after 24 (T24) and 48 h (T48), as shown in Figure 7.

THP-1-like macrophages were infected at MOI 1, based on the previous results obtained with cytotoxic assays, where more differences among strains were observed at lower concentrations (Figure 6a,c). Despite our environmental strains showing a different capacity to enter host cells, with average cells recovered at T0 ranging from 2.24 (ST3130) to 4.33 (ST445) Log CFU/well, this was not significantly different from the *L. pneumophila* reference strain NCTC 12821 (average of 3.04 Log CFU/well), with the only exception of ST2617 (4.19 Log CFU/well at T0). On the other hand, some environmental strains showed a limited intracellular replication capacity when compared to the reference strain. In particular, lower bacterial loads were observed at T24 for ST1 (5.42 Log CFU/well), ST445 (5.42 Log CFU/well), ST1324 (4.65 Log CFU/well), and ST2617 (5.4 Log CFU/well); and at T48 for ST445 (6.4 Log CFU/well), ST579 (6.43 Log CFU/well), ST1324 (6.27 Log CFU/well), and ST2617 (6.29 Log CFU/well).

Pearson correlation coefficients were computed to search for correlations between replication ability, expressed as Δlog10 CFU at 24/48 h, and virulence-associated genes and classes (Appendix A, respectively). A positive, although non-significant, correlation between Dot/Icm (type IVB) effectors and intracellular replication could be noted.

## 4. Discussion

*Legionella pneumophila* is an opportunistic pathogen responsible for Legionnaires’ disease. Most clinical cases are linked to *L. pneumophila* strains inhabiting anthropogenic water systems, which are the primary source of infection [7,45]. Colonization of water systems often involves different *Legionella* populations, whose composition may vary due to several factors (temperature, biofilms, material of the pipes, presence of protozoa, bacterial community composition) [1]. Hence, *Legionella* contaminations can be characterized by a broad genomic diversity, comprising strains with a wide range of virulence genes [46]. Comparative genomic analysis highlighted differences in the virulence gene-related content of *Legionella* retrieved from water samples [46,47], while functional experiments on cellular models were performed to define distinct virulence phenotypes regarding specific steps of the infection process [12,30,48]. In this study, 13 environmental *L. pneumophila* strains, representative of contaminations found in drinking-water systems and cooling towers, were investigated by WGS for their genomic virulence potential and by macrophage-based infection assays to determine strain-specific pathogenic properties.

During a six-year environmental surveillance of *Legionella* in building water systems, we identified several *L. pneumophila* contamination events characterized by variable concentrations across a broad temperature range. The observed heterogeneity underscores the ability of *L. pneumophila* to survive and proliferate under diverse conditions, consistent with its recognized ecological adaptability [49]. The collection of environmental *L. pneumophila* isolates obtained through the monitoring activity also showed a high diversity in terms of serogroups and STs, including clinically relevant ones. Sg1 was the most frequently isolated. This serogroup is considered the principal etiological agent of LD [17] and its presence in drinking water systems has been widely reported [50,51]. Sg6 and sg8, often associated with LD outbreaks, were also found [52,53]. Among the strains identified through SBT, ST1 was the one most frequently associated with contamination cases. The high distribution of ST1 observed in our samples is consistent with its ability to adapt and survive in anthropogenic water systems, which makes it the most widespread *L. pneumophila* strain and the main causative agent of LD [54,55]. Conversely, the presence of ST445, the second most represented strain in our collection and detected across all three systems, has so far been less described in environmental samples. ST1324, one of the most abundant STs found in our contamination cases, has already been reported in both clinical and environmental samples from different countries (Canada, Italy, Greece, China, and Japan) [56,57,58,59]. Similarly to ST445, ST356 and ST1358 were well represented in our collection, despite their occurrence in plumbing systems being is less documented. Within the STs with a sporadic occurrence, i.e., those found in no more than two contamination cases, only ST22, ST23, and ST579 have already been reported for contaminating different type of samples, including building water systems [56,60,61]. Compared to other studies covering broader geographical areas, the genetic diversity reported in this work is noticeably high [58,62]. Indeed, this evidence is particularly relevant from a local perspective, considering that all *L. pneumophila* strains were found in buildings of the same University located in a single city. The STs identified in our study were isolated from three different buildings’ water systems. Within hot water production systems, the highest strain diversity was found among the single-point boilers, suggesting that each of these systems may feature peculiar environmental conditions affecting *Legionella* proliferation [63]. In the cooling tower, we found a broader *L. pneumophila* strain diversity, with 4 different STs, 2 of which (ST579 and ST1358) were exclusively present in this site. Indeed, compared to hot water production systems, cooling towers operate under distinct physical, chemical, and microbiological conditions, as they are based on non-potable water intended for industrial applications, namely the building’s air conditioning. Together, these factors may promote the presence of diverse *L. pneumophila* strains and contribute to the detection of specific STs [59,64]. Overall, the broad genetic variability of *L. pneumophila* contaminations, highlights the adaptability of different strains to diverse environmental contexts, while also suggesting that the characteristics of each water system, even in a limited geographical area, play a key role in shaping the extent and structure of this diversity.

A set of 13 environmental *L. pneumophila* strains, each representing a different ST of our collection, was selected for WGS analysis to explore their genomic relationships and content. The genetic diversity of the *L. pneumophila* isolates was confirmed by cgMLST analysis, as indicated by their allelic distances. Notably, the isolates grouped within the same CC in the MST-SBT analysis also exhibited the smallest number of allelic differences in the core-genome comparison, supporting their close phylogenetic relationship. On the contrary, the high number of singletons highlighted the overall heterogeneity of the strain collection recovered during environmental surveillance. The accessory genome revealed further variability, with gene clusters associated with diverse functions being unevenly distributed among the strains, along with several strain-specific genes. Such differences could be responsible for functional variability and may confer distinct phenotypic traits, including potential variations in virulence and environmental adaptation. As most of the GCs of the accessory genome, including singletons, are associated with virulence and defense mechanisms, we performed a comparative analysis of the virulence gene content to further explore this variability. The VFDB revealed a distinct distribution pattern of such genes. Core *Legionella* effectors were conserved across all isolates, consistent with their established roles in adhesion and intracellular survival. In contrast, effectors involved in vesicle trafficking, non-canonical ubiquitination, and actin dynamics, displayed a variable presence among the strains, potentially influencing host–pathogen interactions [65]. Variation was also detected in the *Lvh locus*. Although the Lvh secretion system was not required for intracellular replication in THP-1 macrophages, strains lacking a complete Lvh *locus* showed a tendency toward lower cytotoxicity, suggesting a potential, albeit inconclusive, role for Lvh in modulating virulence under the tested conditions. Previous studies have indeed linked Lvh to virulence-related phenotypes, particularly under environmental conditions, but have shown that it is not required for intracellular growth at 37 °C [66]. Interestingly, only two STs (ST596 and ST3130) showed capsule synthesis genes. While these annotations point to rare gene acquisitions, possibly through horizontal transfer or functional convergence, their biological relevance in *L. pneumophila* remains uncertain. Such sporadic findings nevertheless further distinguish these sequence types from the reference panel and underscore the genomic plasticity of environmental isolates. Overall, our genomic analysis shows that *L. pneumophila* possesses a highly open pangenome dominated by a large and diverse accessory genome. Many functions relevant to environmental adaptation and virulence, including Dot/Icm effectors and Lvh-associated *loci*, fall within the shell and cloud genome rather than the core. This indicates substantial genomic plasticity among environmental isolates driven by recombination and horizontal gene transfer [25,67,68], and provides a genomic basis for the strain-specific phenotypic differences observed in our macrophage assays.

To further investigate the diversity of the retrieved *L. pneumophila* strains, we evaluated their virulence potential in THP-1-like macrophages, a human alveolar macrophage cell line widely employed to study *Legionella* pathogenetic mechanisms [12]. When comparing the cytotoxicity of *L. pneumophila* STs, we observed significant differences in the survival of the host, also affected by infection time and MOI. Overall, we observed that ST23, ST579, ST1541, and ST1904 were the least effective in inducing cell death. On the contrary, ST22 and, above all, ST596 and ST1324 were associated with the highest levels of cytotoxicity. Our data are partially congruent with information reported in the literature. In fact, while strains such as ST22 and ST1324 have been linked to clinical LD cases [69], ST596 was never reported as a disease-associated strain. On the contrary, although ST23 did not exhibit strong virulence features in our assay, this strain is included in the top five LD-associated STs, accounting for nearly half of the unrelated cases of the disease worldwide [68,69,70]. Similarly, ST1, despite being the most associated with LD outbreaks [71], was not among the strains showing the strongest cytotoxicity. On the other hand, ST1541 and ST1904 have not been linked to LD disease. This discrepancy may be explained by the different distribution of environmental strains and their persistence in water systems. Hence, virulence potential should be considered a key factor in risk assessment along with additional features such as adaptability to environmental conditions and persistence in water systems [46].

Beyond strain-specific factors, bacterial concentration also plays a crucial role in determining pathogenic outcomes. After 48 h of infection and at an MOI of 10, the tested strains were nearly equivalent in inducing cell death. Sousa et al. suggested that there is a threshold above which *L. pneumophila* strains show similar ability to affect host death [48]. This apparent loss of differential cytotoxicity may reflect a shift from strain-dependent pathogenic interactions to global cytotoxicity imposed by excessive bacterial burden. Indeed, at high MOI, macrophages are simultaneously exposed to overwhelming Type 4 Secretion System-dependent effector delivery and high levels of pathogen-associated molecular patterns (PAMPs). These factors can trigger non-specific inflammatory and pyroptotic responses and generate generalized host-damage mechanisms rather than strain-associated virulence profile. Hence, the risk to public health linked to exposure to *Legionella* is not solely related to the presence of specific virulence phenotypes, but also to the bacterial load in the water system. This reinforces the need to manage *Legionella* concentration in artificial water environments to prevent outbreaks.

Intracellular replication also represents a pathogenic mechanism of *Legionella pneumophila.* All environmental *L. pneumophila* strains tested here were able to enter and replicate within THP-1-like macrophages. However, strain-dependent differences in replication dynamics were observed. Among all the STs, ST2617 showed a higher efficiency in macrophage entry, despite its bacterial load being lower than that of NCTC 12821 at 24 and 48 h post-infection, suggesting that efficient entry does not necessarily correlate with sustained replication. In contrast, strains ST1, ST445, and ST1324 exhibited reduced intracellular growth after 24 h, indicating a slower replication rate or delayed adaptation to the intracellular environment. Overall, most environmental strains displayed growth kinetics comparable to those of the reference strains, yet the observed heterogeneity highlights the variable intracellular behavior among *L. pneumophila* populations. Such diversity is consistent with previous studies describing inter-strain variability in intracellular replication capacity. For instance, Nishida et al. reported an environmental isolate unable to replicate within host cells [30], while Gomez-Valero et al. demonstrated marked differences in replication among *Legionella* species, classifying *L. pneumophila fraserii* as non-infectious after 72 h [24]. Conversely, our results contrast with those of Tachibana et al., who found that isolates from hot spring environments exhibited enhanced intracellular replication relative to the reference strains [72].

In addition to the qualitative comparison of virulence repertoires and infection phenotypes, we also explored whether variation in virulence-associated genes could underlie the strain-dependent differences observed in cytotoxicity and intracellular replication. Correlation analysis suggested two biologically plausible trends: strains encoding a higher number of Dot/Icm (type IVB) effectors tended to exhibit stronger intracellular replication, whereas the presence of Lvh (type IVA) secretion system genes showed a negative association with macrophage survival, consistent with a higher cytotoxic potential. These correlations did not remain significant after multiple-testing correction due to the limited number of strains and are therefore interpreted strictly as descriptive. Nonetheless, the observed patterns are coherent with known functional distinctions between Dot/Icm and Lvh systems and may partially explain the phenotypic heterogeneity observed among environmental isolates.

The use of THP-1-like macrophages as an intracellular infection model represents a limitation to this study, even though they are extensively used in infection assays. Indeed, in vitro studies do not reflect the complexity of pathogen–host interactions, only showing the virulence potential of bacterial strains regarding specific steps of the infection cycle. Moreover, the redundancy of virulence effectors encoded in the *Legionella* genome means overlapping in protein function. Thus, the presence or absence of single effectors may not directly influence the pathogenic properties observed experimentally.

## 5. Conclusions

Our study highlighted the genomic and functional diversity of *L. pneumophila* environmental strains isolated from distinct water systems of academic buildings. We observed a high genetic heterogeneity at both the phylogenetic and gene content levels, even though all the strains investigated originated from a very limited geographical area as they were isolated from university buildings located in a single city. Furthermore, strain-specific differences were observed in inducing host cell death and, to a lesser extent, intracellular replication capacity. Differences in the genetic repertoires, cytotoxicity, and intracellular behavior reflect strain-specific virulence potential, suggesting the presence of distinct pathogenic strategies within the same *Legionella* species. These findings support the need for a comprehensive environmental surveillance and risk assessment, as the potential for infections may be affected by the composition of *L. pneumophila* populations. Therefore, an understanding of the diversity of environmental *Legionella* strains is fundamental for preventing LD and managing contamination in water systems.

## Figures and Tables

**Figure 1 microorganisms-13-02832-f001:**
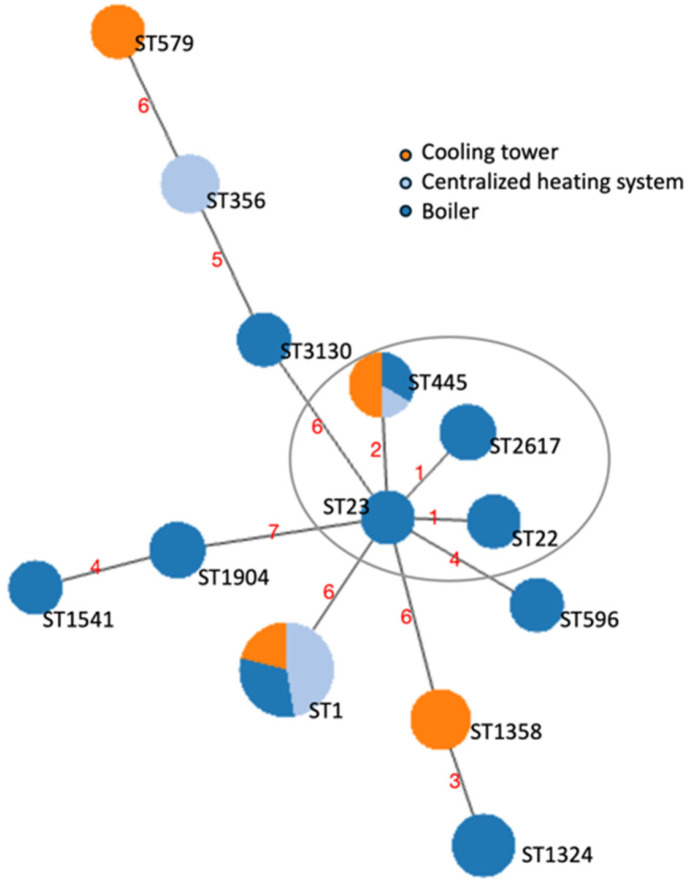
Minimum spanning tree (MST) of the sequence types (STs) identified through the sequence-based typing (SBT) of the 47 *L. pneumophila* strains isolated from contaminated water samples. The nodes represented as circles correspond to the STs found. The size of the circles is representative of the number of cases, and the color reflects the contaminated water system of origin. The length of the branches is proportional to the allelic differences among strains, displayed by red numbers. The STs that differed for one *locus* (single-locus variants) or two *loci* (double-locus variants) are considered part of a Clonal Complex (CC) and are circled in grey.

**Figure 2 microorganisms-13-02832-f002:**
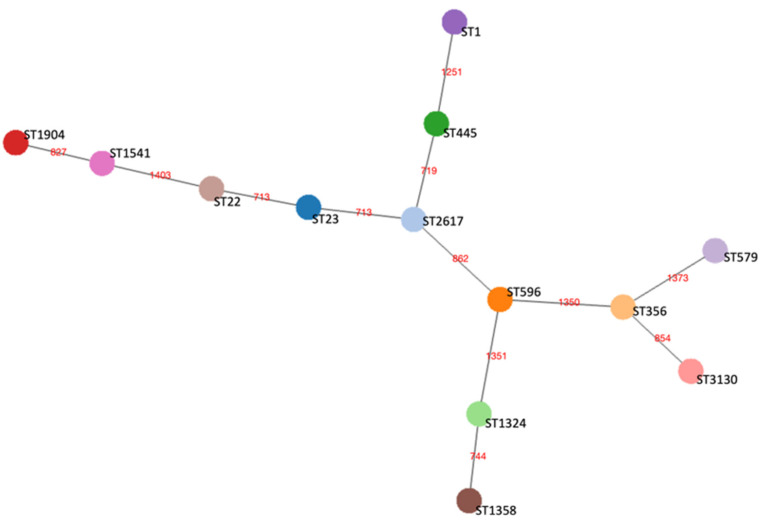
Core genome minimum spanning tree (cgMST) of the 13 representative *L. pneumophila* strains belonging to each ST based on a set of 1411 core genome Multilocus Sequence Typing (cgMLST) target genes. Nodes represented by circles of different colors display STs and are connected by branches labelled with red numbers indicating the allelic distance between strains.

**Figure 3 microorganisms-13-02832-f003:**
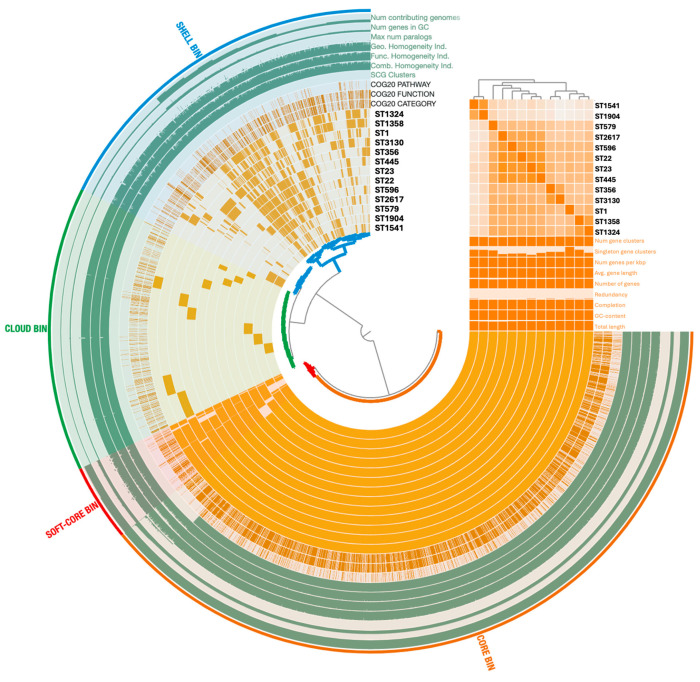
Pangenome distribution across the 13 *L. pneumophila* strains representing each ST. The analysis was performed using the anvi’o 8 pangenome pipeline. Arcs correspond to the genomes analyzed and organized on the Gene Clusters (GCs) frequencies, while orange blocks represent GCs. Genomic features (e.g., single-copy gene (SCG), Clusters of Orthologous Groups (COG), GC content) are displayed in additional layers. Statistics about the pangenome are reported as histograms, and Average Nucleotide Identity (ANI) as a matrix. Core and accessory genomes are highlighted with colors.

**Figure 4 microorganisms-13-02832-f004:**
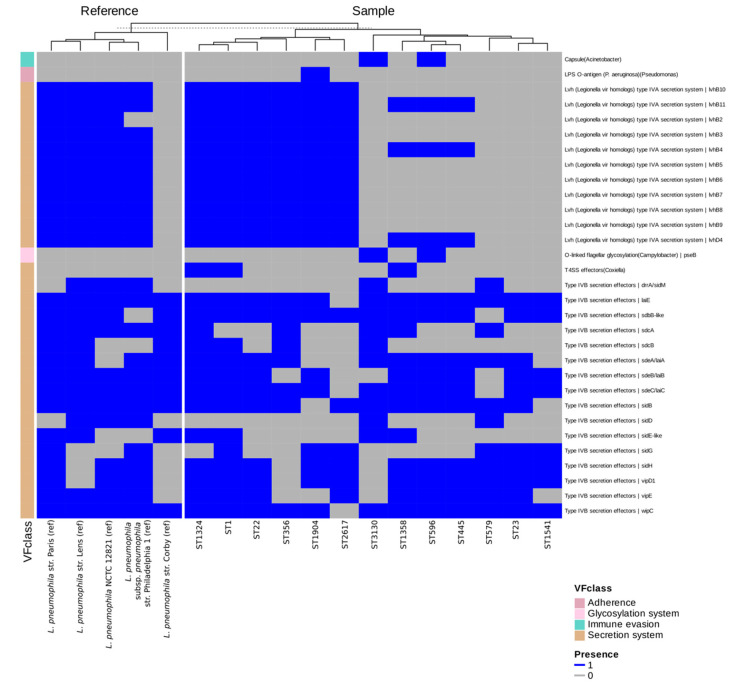
Heatmap showing the presence (blue) and absence (grey) of virulence determinants across the genomes of the 13 *L. pneumophila* strains representing each ST and reference genomes provided by the VFDB database. Only variable determinants are displayed for clarity, including subsets of Dot/Icm (type IVB) secretion system effectors and the Lvh (type IVA) secretion system. Conserved factors are omitted from the figure but listed in Appendix A, which provides the complete presence–absence dataset. STs are indicated on the *x*-axis, while virulence factors are grouped by functional class on the *y*-axis. Isolates and reference strains were clustered according to their virulence factor presence–absence profiles using hierarchical clustering, resulting in grouping of strains with similar repertoires.

**Figure 5 microorganisms-13-02832-f005:**
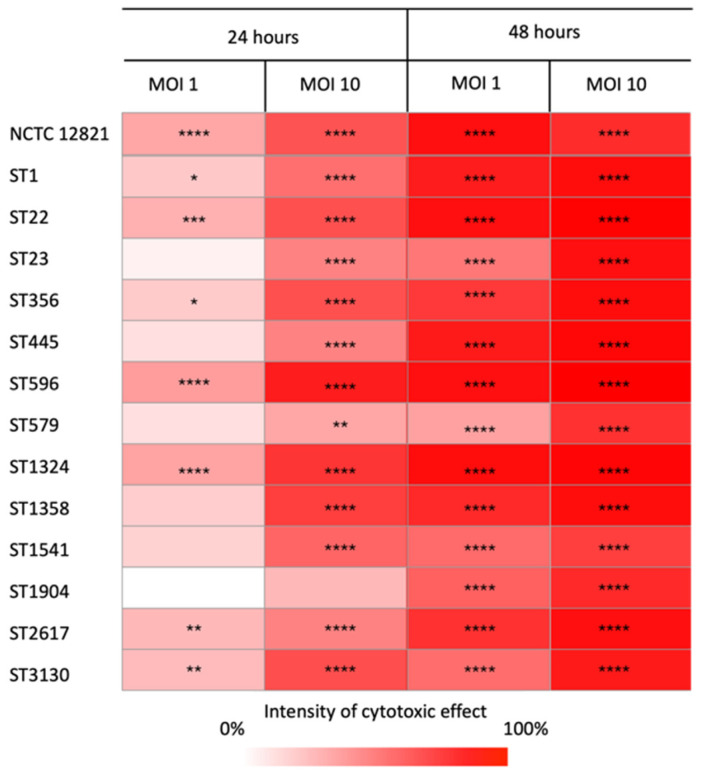
Cytotoxicity matrix displaying the ability of the representative isolates of each ST to induce macrophage cell death compared to the control. The percentage of the strains’ cytotoxicity was calculated as the reduction in THP-1 cell viability (mean ± standard error) relative to the control. The color intensity reflects the killing ability of the strains, with darker shades indicating higher host cell death at each multiplicity of infection (MOI) and time. Differences in survival rates were considered significant if * *p* ≤ 0.05; ** *p* ≤ 0.01; *** *p* ≤ 0.001; **** *p* ≤ 0.0001.

**Figure 6 microorganisms-13-02832-f006:**
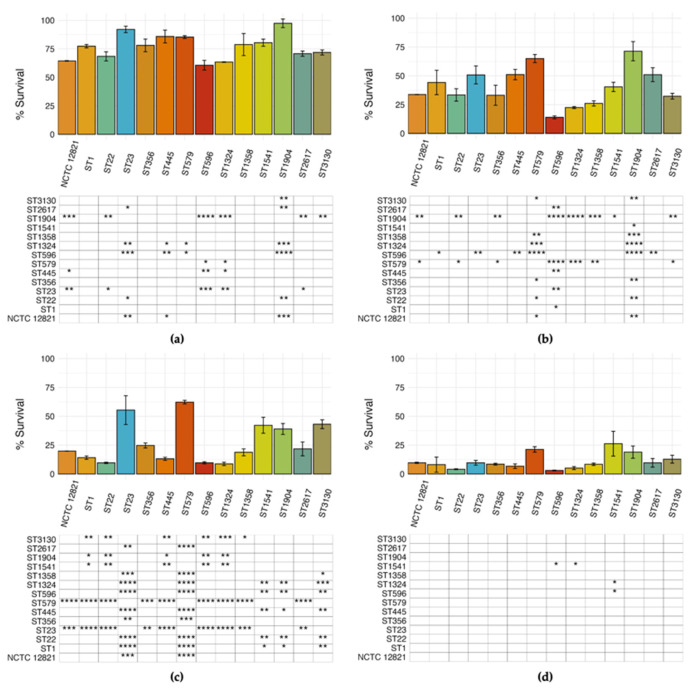
Relative survival of macrophages infected by the representative *L. pneumophila* STs and NCTC 12821 reference strain after 24 h at (**a**) MOI 1 and (**b**) MOI 10 and 48 h at (**c**) MOI 1 and (**d**) MOI 10. The percentage of surviving macrophages was calculated as the percentage of viable macrophages compared to the control. Differences detected by the ANOVA test were considered significant when * *p* ≤ 0.05; ** *p* ≤ 0.01; *** *p* ≤ 0.001; **** *p* ≤ 0.0001 and displayed in the matrices below the chart after Tukey’s HSD pairwise comparison.

**Figure 7 microorganisms-13-02832-f007:**
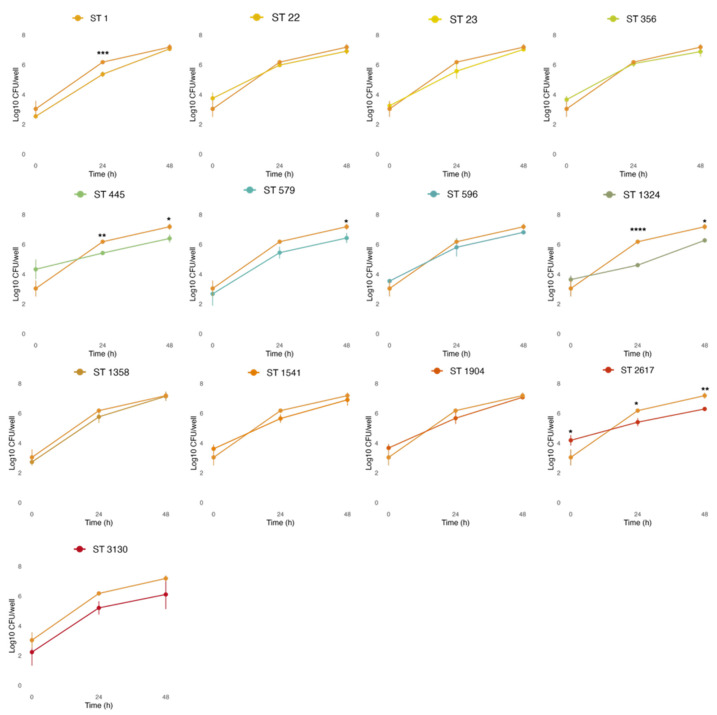
Intracellular growth of the STs compared to the reference strain *L. pneumophila* NCTC 12821 (light orange line). THP-1 cells infected at an MOI of 1 were lysed, and serial dilutions were used to enumerate the *L. pneumophila* cells released on BCYE at 0, 24, and 48 h post-infection. All values represent the average and the standard deviation for three identical experiments. Differences in the bacterial load at each time point were considered significant (* *p* ≤ 0.05; ** *p* ≤ 0.01; *** *p* ≤ 0.001; **** *p* ≤ 0.0001).

**Table 1 microorganisms-13-02832-t001:** Sequence types (STs) identified in *Legionella pneumophila* (*L. pneumophila*) contamination cases from different water systems over six years of monitoring activity.

ST	Allelic Profile	Sg	Number of Cases ^1^	Water System ^2^	Year	Water Temperature (C°) ^3^	Contamination Level (CFU/L) ^4^
1	1,4,3,1,1,1,1	1	19	C, B, CT	2015, 2016, 2017, 2018, 2019, 2020, 2021	34 (12.8–61)	2.69 × 10^4^ (1 × 10^2^–4.25 × 10^5^)
22	2,3,6,10,2,1,6	1	1	B	2017	22.7	1 × 10^2^
23	2,3,9,10,2,1,6	1	1	B	2017	53	7.4 × 10^3^
356	6,10,15,12,9,14,11	1	4	C	2015, 2017, 2019, 2020	37.78 (22.4–50.1)	1.3 × 10^3^ (1 × 10^2^–3 × 10^3^)
445	2,3,18,13,2,1,6	1	6	C, B, CT	2016, 2017, 2018, 2020, 2021	31.69 (19.7–38.1)	3.19 × 10^4^ (1 × 10^2^–9.5 × 10^4^)
579	3,13,1,3,14,9,11	1	1	CT	2017	25	9.5 × 10^4^
596	27,3,9,43,56,5,6	6	1	B	2019	36.5	2.4 × 10^3^
1324	5,1,22,30,6,10,203	8	5	B	2015, 2016	29.98 (16.7–41)	2.67 × 10^4^ (1 × 10^2^–8.3 × 10^4^)
1358	5,2,22,10,6,25,203	8	4	CT	2017, 2018, 2019, 2020	30.67 (28.2–34.3)	1.1 × 10^4^ (3 × 10^2^–2.3 × 10^4^)
1541	21,27,28,54,15,29,206	4	1	B	2017	33.9	1.4 × 10^4^
1904	21,14,29,16,15,29,22,7	7	2	B	2020, 2021	31.05 (30.2–31.9)	1.27 × 10^4^ (3 × 10^3^–2.25 × 10 ^4^)
2617	2,3,9,10,2,1,209	8	2	B	2017, 2019	32.5 (19–46)	4.89 × 10^4^ (2.6 × 10^3^–9.5 × 10 ^4^)
3130	6,10,19,28,2,4,3	6	1	B	2021	41.3	6 × 10^4^

^1^ Number of contaminations associated with each ST. ^2^ Origin of each ST. C, centralized heating system; B, boiler; CT, cooling tower. ^3^ Average temperature of the water samples from where STs were isolated. Minimum and maximum values are displayed in brackets. ^4^ Average of the total contamination levels associated with each ST, regardless of water system and sample. Minimum and maximum values are displayed in brackets.

**Table 2 microorganisms-13-02832-t002:** General genomic characteristics determined by QUAST and CheckM2 analyses of the *L. pneumophila* strains sequenced in this study.

Genomic Feature	Average Value	Value Range
Genome length (Mb)	3.4	3.2–3.6
Number of contigs	46.54	26–67
GC content (%)	38.1	38.11–38.28
N50 (Mb)	0.2	0.07–0.5
Completeness (%)	100	100
Contamination (%)	0.13	0.02–0.26

## Data Availability

The original contributions presented in this study are included in the article/Appendix A. Sequence data are available on NCBI under the accession numbers from JBRULC000000000 to JBRULO000000000. Further inquiries can be directed to the corresponding author.

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
