# Peer review of "Unravelling the Genomic and Virulence Diversity of *Legionella pneumophila* Strains Isolated from Anthropogenic Water Systems"

_microorganisms, 2025, doi:10.3390/microorganisms13122832_

Round 1

Reviewer 1 Report

Comments and Suggestions for Authors

The authors present herein the results of genomic analysis of different Lp isolates over a 6-year period. Furthermore, the authors tried to investigate potential connections between virulence and different genotypes. The manuscript is in general well prepared and well presented. The experiments done are robust and the conclusions drawn are to the point. Perhaps the authors could take advantage of the results that they have presented and prepare some more graphs with potential interconnections among virulence/replication capability/STs, etc. Attached they will a pdf file with some comments that they may find useful.

Reviewer 2 Report

Comments and Suggestions for Authors

GENERAL COMMENTS:

This article by Barigelli et al. reports the genomic characterization of Legionella pneumophila strains isolated from potable water systems and one cooling tower in a university campus.  Also, the authors conducted a cytotoxicity assay with macrophages for representative strains isolated.

The study overall is well conducted and designed, and could be useful to the Legionella community.  However, the main conclusion of the study is not novel because it is already known that Lp strains are highly diverse from a genetic standpoint.  What the authors could do to increase the relevance of their study, is to focus on their particular results comparing with previous reports and discussing their results by STs according to abundance.  Figure 1 is an excellent reference for such discussion, since the authors state that the size of the circles is proportional to the abundance of the corresponding ST (number of cases).  For instance, the discussion could be subsectioned for the different STs in order of abundance.  Also, I suggest that the STs exclusively present in the cooling towers should be discussed separately to contrast the strains present in potable water systems vs strains in industrial-type water (that is, non-potable water).

Consequently, I suggest to the authors to re-write their discussion to strengthen the particular relevance of their results from a local perspective.  In this respect, it should be clearly stated that the samples used come from a very local environment (all being collected from a single university campus).

One general comment about the Figures, is that they are too small, (at least the font should be increased).

SPECIFIC COMMENTS BY LINE NUMBER

Line 74. Please specify in the text that reference #18 primarily concerns hot water environments in hotels.

Line 97. Reference #26 is from 2014. If no recent references exist to substantiate the statement made, perhaps the authors should address this as a need.  However, if more recent references exist, please cite them here.

Line 109. From a strictly technical standpoint (by definition), Lp's life cycle does not include macrophages, as the evolution of the species has naturally depended on interactions with amoeba.  Please change "life cycle" for 'replication'.

Line 112. Please specify that the academic buildings are from a single university campus.

Line 119. Please change "intracellular cycle" for 'intracellular growth cycle'.

Line 126. In relation to the cooling tower, what tank is being referred here? Please clarify by mentioning what type of cooling tower was sampled, the volume of water sampled, the type of disinfectants/biocides used in the cooling tower.  Also, please mention whether or not there are other cooling towers in campus, and if so, how representative of the other cooling towers is the one sampled?

Lines 135-136. It is not clear how a presumptive colony was regarded as representative.  Please clarify the method by which the number of representative colonies was determined, and which colonies were used to estimate the levels of Legionella in the water sampled.

Line 141. In relation to section 2.3, were the sequences obtained submitted or stored in a public database?  Please explain and detail.

Line 195 = Section 2.5, and line 217 in particular. Explain how the viability of macrophages was determined using Alamar blue, and how the numbers were handled to determine the level of cytotoxicity.

Line 202. Were the Lp strains cultured on BCYE plates as lawns or as isolated colonies? If cultured as lawns, 48 h lawns could be considered as still replicating, and replicative Lp is less infectious than stationary phase Lp.  Please clarify.

Lines 207-208. Technically speaking, an OD of 2.5 units cannot be measured.  I am assuming that the OD was measured in a spectrophotometer or a plate reader using dilutions of the bacterial suspensions.  Also, how did the authors determined that an OD of 2.5 units correspond to 2.2 x 109 bacterial cells?  Please clarify/explain.

Line 213.  Were controls run to confirm that strains were sensitive to 0.1 ug/ml of gentamicin?

Lines 289-291. The text contained within these lines should be deleted.

Lines 356-357.  Please delete the phrase "representing the ratio of the bacterial and cellular concentrations".

Comments on the Quality of English Language

The English Language of the paper is quite good.  Only a few typos and grammatic issues were detected, some of which are mentioned below.

Line 148. Add the word "using" before "...the seven loci standard protocol established by the European..."

Line 163. Change "Dna" for 'DNA'.

Lines 172-173. Move "(v.39.28) [35]" from line 173 to line 172 right after "BBTools".

Lines 235, 248, 354,381, 400, 407. Put L. pneumophila in italics.

Lines 235-237. The wording of the sentence is not clear. Please re-write.

Lines 292, 321, 396.  The "P" of Pneumophila needs to be lower case.

Reviewer 3 Report

Comments and Suggestions for Authors

The authors conducted a detailed genomic characterization study of Legionella isolates obtained from water sources in academic centers and further assessed their virulence potential through experimental cell culture assays. The topic is highly relevant, as it addresses a microorganism that remains relatively underexplored but has significant pathogenic potential and public health importance. The authors provided an excellent characterization of the isolates and presented a solid experimental approach.

I found no methodological or scientific writing flaws. The overall presentation and manuscript writing are of high quality.

I only have minor revisions to suggest and would like to congratulate the authors for the well-designed study and carefully written manuscript.

Minor revisions

  • In the title, italicize the scientific name. Double-check the use of italics for all mentions of the microorganism throughout the manuscript.
  • In the keywords, replace terms already included in the title to improve indexing.
  • L231: use pneumophila starting with a lowercase letter. Verify consistency across the manuscript.
  • L461: revise to “A set of 13 environmental L. pneumophila strains.”

Reviewer 4 Report

Comments and Suggestions for Authors

The current research article “Unravelling the genomic and virulence diversity of Legionella pneumophila strains isolated from anthropogenic water systems” demonstrates the comparing the genomic diversity of Legionella pneumophila strains isolated in water systems of academic buildings, together with their cytotoxicity and intracellular replication in THP-1-like macrophages. The obtained results are quite interesting and significant both from fundamental and practical point of view. However, there are several questions and comments which should be addressed during the major revision.

  1. Line 19. I do not agree with this claim. Are there any references?
  2. Line 28. Please, add the range of genome size.
  3. Why did you need sequencing coverage normalization? You sequenced not metagenome.
  4. Line 176. Authors did not use the CheckM2? It is more accurate and confident.
  5. For each 13 sequenced genomes should be provided genomic features, including contig counts.
  6. Were the genomes assembled into single ring of chromosome?
  7. What about plasmids? Were there any of them?
  8. What is main conclusion from pangenome analysis of L. pneumophila?
  9. Were all 13 strains assigned to Legionella pneumophila by genome based taxonomy? Please, check this, using for example GTDB toolkit.
  10. How readers can get something from Figure 3? Maybe there should be some indicators, e.g. strain number and etc.

Reviewer 5 Report

Comments and Suggestions for Authors

In their manuscript, Barigelli et al. present a comprehensive genomic and functional characterization of Legionella pneumophila strains isolated from anthropogenic water systems across a six-year surveillance period. Using SBT, cgMLST, pangenomics, virulence gene profiling, and macrophage infection assays, the authors convincingly demonstrate that environmental L. pneumophila populations are genetically heterogeneous and exhibit marked, strain-specific cytotoxicity and intracellular replication phenotypes. The manuscript is clearly written, methodologically sound, and provides valuable insight for environmental surveillance and risk assessment of L. pneumophila. I find the work robust and impactful. I recommend its acceptance in the present form, pending only very minor editorial adjustments.

Minor:

Different STs show markedly distinct cytotoxicity and replication phenotypes, yet the manuscript does not explore whether any specific virulence gene modules (e.g., variable Dot/Icm effectors, presence/absence of Lvh, or accessory genes from the shell/cloud genome) are statistically associated with the observed phenotypes. Could the authors comment on whether any genotype–phenotype trends were detectable, even if only at the descriptive level.

Figure 6 shows that strain-specific cytotoxicity differences are highly pronounced at MOI 1 but largely disappear at MOI 10. Could the authors comment on the biological mechanism underlying this dose-dependent convergence? Does this reflect a saturation effect in macrophage killing, or could high bacterial load activate non-specific cell death pathways?

Some strains lack the Lvh (type IVA) secretion system while all strains retain the Dot/Icm (type IVB) system. Could the authors elaborate on the functional differences between these two secretion systems and whether the absence of Lvh is expected to impact the phenotypic outcomes observed in this study?

Line 413: “6.27 CFU/well” → missing “Log”.

Round 2

Reviewer 4 Report

Comments and Suggestions for Authors

No further comments.